# An Ingested Orthodontic Wire Fragment: A Case Report

**DOI:** 10.3390/dj4030024

**Published:** 2016-08-01

**Authors:** James Puryer, Catherine McNamara, Jonathan Sandy, Tony Ireland

**Affiliations:** 1School of Oral and Dental Sciences, Bristol Dental Hospital, Lower Maudlin Street, Bristol BS1 2LY, UK; jonathan.sandy@bristol.ac.uk (J.S.); tony.ireland@bristol.ac.uk (T.I.); 2HSE Regional Orthodontic Department, St. James’s Hospital, Dublin 8, Ireland; catherinem.mcnamara1@hse.ie

**Keywords:** orthodontic, wire, fragment, ingested

## Abstract

Accidental ingestion or inhalation of foreign bodies has been widely documented, including incidents which occur whilst undertaking dental treatment. Most ingested objects pass through the gastrointestinal tract (GIT) spontaneously, but approximately 10%–20% need to be removed endoscopically and 1% require surgery. This case reports a complication arising from the accidental loss of an archwire fragment during maxillary archwire placement. It describes the immediate and subsequent management, including the use of radiographs to track the passage of the fragment through the gastro-intestinal tract. This case stresses the vigilance that dentists must take to prevent inhalation or ingestion of foreign bodies and the consequences of time-delays when management decisions are needed.

## 1. Introduction

Accidental foreign body aspiration and/or ingestion are well-described phenomena, respected for their serious health risks, with ingestion occurring more frequently than aspiration [1,2]. In the United States, approximately 1500 people die each year following ingestion of foreign bodies in the upper gastrointestinal tract [1], with the most frequently ingested foreign objects including coins, meat boli, button batteries, and dental objects [1,3]. Most ingested objects pass through the gastrointestinal tract (GIT) spontaneously, but approximately 10%–20% need to be removed endoscopically and 1% require surgery [1].

The most frequently aspirated foreign objects are organic objects such as peanuts, beans and sunflower seeds [2]. Such incidents usually occur in children less than 3 years of age, with a male: female ratio of 1.2:1 [2]. Some individuals are at greater risk of ingesting and/or aspirating a foreign object. This includes those with an accompanying medical, mental or physical disability where there is a significant increasing risk in these children up to 15 years of age [1,4]. Adult high-risk groups include: patients with mental disability, dementia, those on prescribed opiate or antidepressant medication, alcoholism, neurological disorders (such as Parkinson’s disease or stroke related dysphagia) and epilepsy [1,2].

The incidence of aspiration or ingestion of foreign bodies of dental origin varies considerably in the literature [1,3]. Reports of such accidents are not common [4] but significantly, foreign objects of dental origin are the second most common source [1,5]. In addition, the incidence of foreign body ingestion or inhalation of dental origin is more common in adults than children [1,3]. Dental objects ingested or inhaled include: toothpicks, files, reamers, burs, impression materials, restorative inlays, crowns, post and cores, onlays, rubber dam clamps, removable prostheses, implant components, dental implant screw drivers, mirror heads and even a tooth during extraction [1,3,5,6,7].

The incidence of reported cases in orthodontics is considerably less common but no less varied in the range of objects involved, and these include brackets, bands, second molar buccal tubes, transpalatal arches, removable appliances and appliance fragments, archwire fragments, sectional archwires, coil springs, expansion appliance keys, retainers, and quadhelices [2,3,4,8,9,10,11,12,13,14,15,16,17,18,19,20,21,22,23,24,25,26,27,28,29]. A literature review was carried out using the PubMed database to search for case reports relating to ingestion or aspiration of foreign bodies of orthodontic origin. The search was carried out using a combination of the keyword “orthodon*” with “ingest*”, “swallow*” and “inhal*”, and these combinations produced a total of 245 results. The titles and abstracts of these 245 papers were then examined. Inclusion criteria were that the papers must be published in English and also report on a case of ingestion or inhalation of a foreign body of orthodontic origin. Twenty five papers that were published between 1983 and 2016 met the inclusion criteria, and these papers reported on 28 individual cases. Ingested objects were reported more frequently, with 22 such cases compared to only 6 reports of inhaled objects. The majority of cases affected children rather than adults, and the majority of cases were related to fixed orthodontic appliances compared to removable appliances. The majority of accidents involved small objects with 8 reported cases involving archwire fragments and 4 reports of accidents involving fixed brackets. However, accidents involving larger objects were also reported including a 3 cm long Kobayashi ligature, a fractured Twin block appliance and an intact quadhelix appliance. There were 4 reported cases of ingested expansion keys. A summary of the papers included in the review is shown in Table 1.

This case reports a complication arising from the accidental loss of an archwire fragment during maxillary archwire placement and the importance of time-delays when making decisions based on time-dependent radiological investigative procedures.

## 2. Case Report

A 15-year-old female patient was undergoing upper and lower fixed appliance treatment, of her Class II division 1 malocclusion. She had no relevant past medical or dental history. During the placement of a rectangular nickel titanium 0.017 × 0.022 inch maxillary archwire, the distal fragment, approximately 1 cm in length, was not retained whilst being reduced with distal-end cutting pliers. The operator noted the flight of the archwire fragment to the oro-pharyngeal region and immediately took actions to retrieve the fragment. The patient was moved from the supine position and thorough oral suction was applied. The patient was asked to cough vigorously and finally to rinse thoroughly. No spontaneous coughing, respiratory difficulty or distress occurred at any time during the incident, and the patient was asymptomatic throughout.

Radiographic investigations were carried out as a matter of urgency and confirmed that the archwire fragment had lodged in the right piriform recess (Figure 1a,b).

The patient was transferred immediately to the Region’s main General Hospital, which was provided with a sample of an equivalent archwire. Approximately three hours later, under general anaesthesia, the patient underwent endoscopic retrieval of the fragment. The archwire fragment could not be found on exploration of the piriform recess, larynx or oesophagus, and further radiographs were taken. It was found that, in the three hour period between the incident and presentation to the General Hospital, the archwire fragment had relocated and had been ingested. Once this was confirmed, no further operative retrieval procedure was pursued, and instead, the patient was monitored. A hospital stay (3.5 days) was necessary due to a post-operative pyrexia and post-instrumentation discomfort. During this in-patient period, serial tracking radiographs confirmed that the orthodontic archwire fragment had passed safely and uneventfully through the gastrointestinal tract (Figure 2 and Figure 3). Following discharge there was rapid recovery to full health. Orthodontic treatment was resumed and completed successfully without further incident.

## 3. Discussion

Ingestion or inhalation of a foreign body is a recognised complication of various dental procedures. Fortunately it is not common. Tiwana et al. [5] found just 36 incidences of dental origin over a 10-year study period. Additionally, most ingested objects pass through the GIT without problems (as in this case report), with 80%–90% passing within 7 to 10 days [1]. The size, shape, and presence of sharp edges of the ingested object will influence risk factors, management and outcome. Large or sharp objects are at risk of becoming impacted, but approximately 60% enter the alimentary canal without lodging in the oesophagus [3]. Additional GIT sites at risk of impaction include the pylorus, appendix, sigmoid colon and anal canal. Perforation may occur, but this is rare [1,3]. Other complications with ingestion of foreign bodies include intestinal mucosal ulceration, obstruction, abscess formation, haemorrhage and fistula formation [4].

Aspiration of a foreign body, during a dental procedure, presents a serious problem [3]. Symptoms will depend on where the object becomes lodged, as well as size and shape. In situations where a large object lies above the vocal cords and obstructs the airway, respiratory distress will occur and will require urgent life-saving action. Small objects tend to pass through the vocal cord area and not give rise to upper airway obstruction, although it is essential that these objects are retrieved too. Whilst initially there may often be no ill-effects or symptoms, long-term pulmonary problems will inevitably arise if the foreign body is left in-situ [2]. Pulmonary abscess formation, pneumonia and bronchiectasis are known to have arisen from unrecognised inhaled foreign objects, with their removal complicated by the formation of granulation or scar tissue [1,3].

Where a foreign object is lost at the back of the mouth during a dental procedure, it is more likely, to enter the gastrointestinal tract, rather than the respiratory tract [4]. The British Orthodontic Society advises that ingested smooth flexible objects less than 5 cm in length are likely to pass through the GIT uneventfully [30], whilst larger objects are more prone to obstruct or perforate the GIT. Removal of these larger objects may be advised. Additionally, more than 50% of foreign bodies will evacuate in the stools unnoticed [1,4]. When an object is accidentally lost irretrievably to the oropharyngeal area, radiologic evaluation is essential to determine if the object has been ingested, aspirated or has become embedded within the deep tissues of the oropharynx [27]. Recommendations on the radiographs that should be taken can be found within the Royal College of Radiologist Guidelines [31]. Bronchoscopy and oesophagoscopy are then the mainstays of foreign object retrieval in the upper airway and aero-digestive tract, though surgical access through the neck, chest or abdomen may be required in certain anatomical circumstances [5].

This case report highlights common issues. The archwire fragment was ingested, rather than aspirated and passed through the GIT without incident. However, at the time of the incident, these two factors could not have been determined or predicted. Immediate retrieval was not possible as the archwire fragment could not be located in the oro-pharyngeal region. Once radiographs confirmed that the archwire fragment was in the piriform recess, retrieval was considered imperative. Initial radiographs, taken approximately 15–20 min after the incident had arisen, suggested that impaction had occurred. The time interval of approximately three hours before the endoscopic retrieval could be undertaken in a General Hospital facility (geographically removed from where the incident occurred) proved significant. Radiographic views confirmed that ingestion of the archwire fragment had occurred in the intervening period.

Accidental ingestion or aspiration of a foreign object is an ever-present risk during all dental procedures, including orthodontic treatment. The majority of orthodontic components are small, and in combination with saliva, can be difficult to handle. In addition, many patients are treated in supine or semi-recumbent position, such that any lost orthodontic component can easily fall into the patient’s oropharynx. Prevention is of primary importance, and preventive measures include: identifying at-risk patients during history taking, ensuring high-speed suction is immediately available at all times, and using appropriate barrier techniques such as rubber dam, gauze [32], or cotton wool rolls. Within orthodontic practice, attaching floss or wax to protect manipulation of small orthodontic components is particularly valuable. Appliances, both fixed and removable, should be retentive. In the case of removable appliances, they should have a radio-opaque component and be smooth, without any sharp edges or clasps. An upright position may be used in preference to a supine position for certain patients and procedures. Long spans of unsupported archwires should be supported with tubing. It is also important to remember that patients wear fixed and removable appliances for long periods of time, and accidental ingestion or inhalation of components can also occur outside of the dental surgery. Thus, it is essential to ensure that any missing appliance or component is accounted for at each patient visit [1,3,11,14]. This case report illustrates the importance of time, both in acting immediately to retrieve the foreign body clinically, and when using radiological data in decision-making. Time delays must be taken into consideration and time-dependent radiographic investigative data used accordingly.

## 4. Conclusions

This is a case report on an ingested orthodontic archwire fragment and describes the immediate and subsequent management. It reinforces that vigilance must be taken by all clinicians when carrying out dental treatment to prevent the inhalation or ingestion of foreign bodies.

## Figures and Tables

**Figure 1 dentistry-04-00024-f001:**
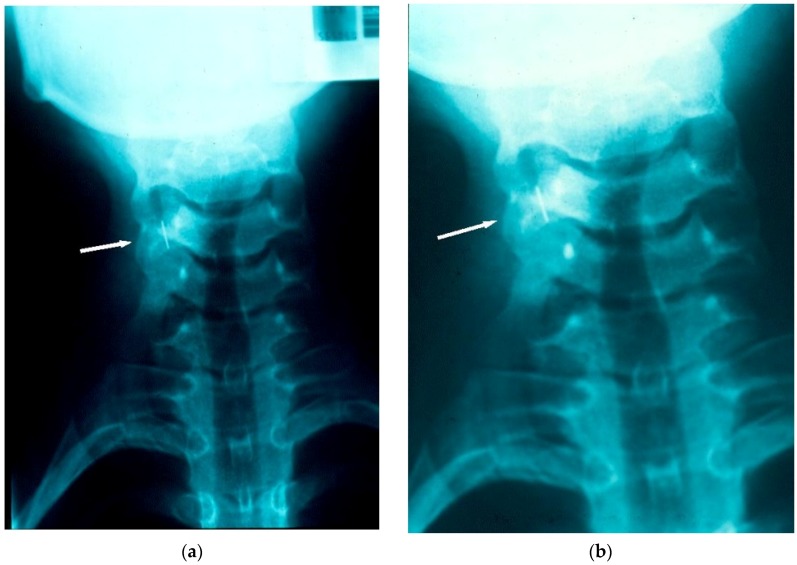
(**a**) Radiograph taken within 15–20 min of the chairside incident, locating the archwire fragment to the right piriform recess; (**b**) Radiograph taken within 15–20 min of the chairside incident, locating the archwire fragment to the right piriform recess.

**Figure 2 dentistry-04-00024-f002:**
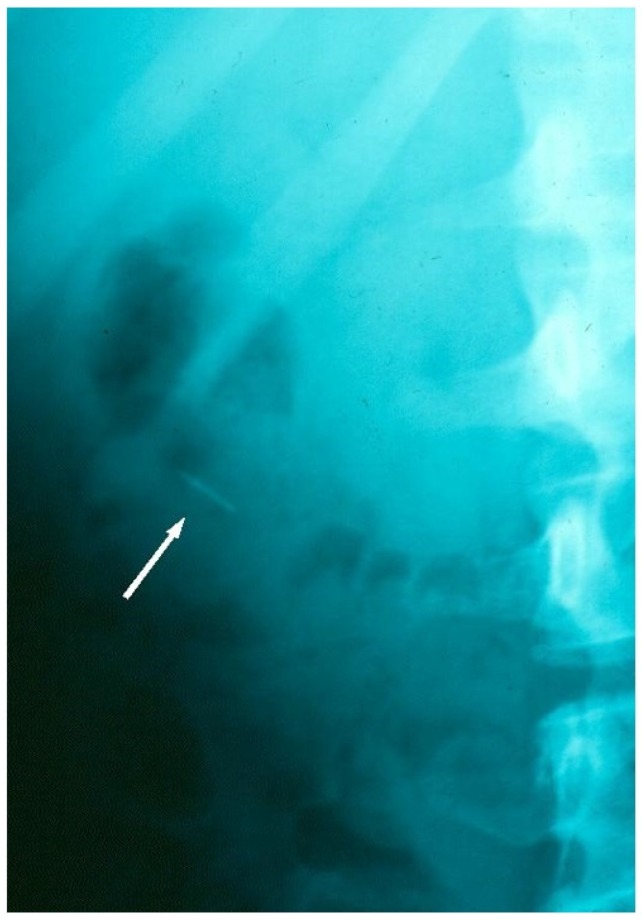
Radiographic tracking view, confirming the archwire fragment location in the colon, during its safe passage through the GIT.

**Figure 3 dentistry-04-00024-f003:**
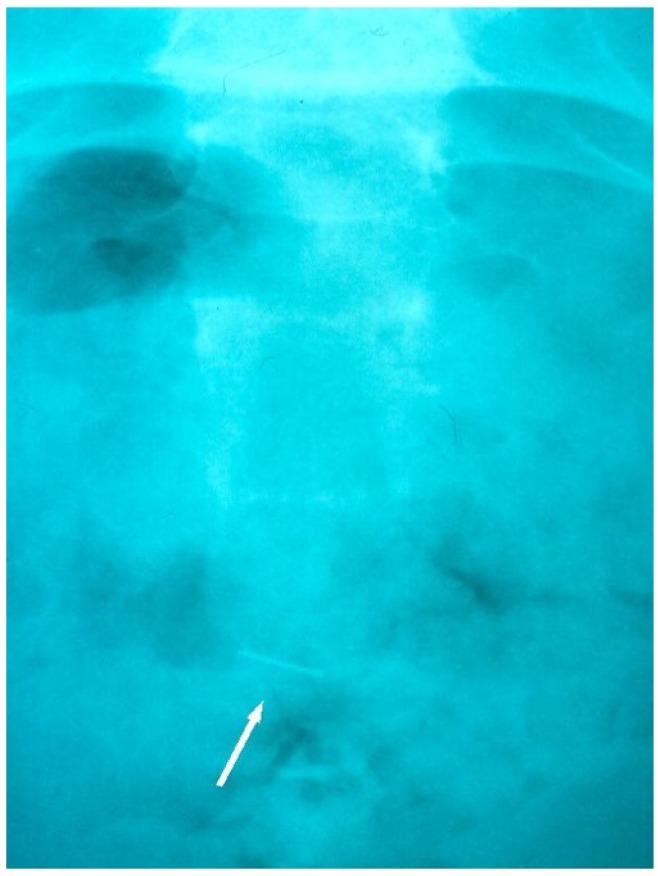
Radiographic tracking view showing the archwire fragment in the rectal area, prior to its natural evacuation from the GIT.

**Table 1 dentistry-04-00024-t001:** The summary of the papers included within the literature review of case studies related to the inhalation or ingestion of foreign bodies of orthodontic origin.

Authors	Year	Foreign Body	Inhaled or Ingested
Wilmott et al. [27]	2016	Fixed bracket	Ingested
Tiller et al. [21]	2014	Ligature wire	Ingested
Hoseini [16]	2013	Orthodontic wire	Ingested
Park et al. [20]	2013	Archwire fragment	Ingested
Naragon et al. [28]	2013	Orthodontic band	Ingested
Umesan et al. [22]	2012	Archwire fragment	Inhaled
Monini Ada et al. [23]	2011	Expansion key	Ingested
Tripathi T et al. [24]	2011	Expansion key	Ingested
Rohida et al. [29]	2011	Twin block appliance	Ingested
Nicolas et al. [9]	2009	Archwire fragment	Inhaled
Sheridan [12]	2009	Fixed bracket	Inhaled
Fiho et al. [13]	2008	Fixed bracket	Inhaled
Allwork et al. [4]	2007	Quadhelix	Ingested
Al-Wahadni et al. [1]	2006	Orthodontic band	Ingested
Abdel-Kader [17]	2003	Transpalatal archwire	Inhaled
Sfondrini et al. [18]	2003	Rapid palatal expander	Ingested
Klein et al. [2]	2002	Retainer	Inhaled
Quick et al. [11]	2002	Wire & coil Spring	Ingested
Milton et al. [8]	2001	Fixed bracket	Ingested
		Archwire fragment	Ingested
		Sectional wire	Ingested
Dibiase et al. [3]	2000	Removable appliance	Ingested
Absi et al. [14]	1995	Archwire	Ingested
Lee [10]	1992	Archwire fragment	Ingested
Parkhouse [25]	1991	Appliance segment	Ingested
Hinkle [19]	1987	Retainer	Ingested
Nazif et al. [26]	1983	Expansion key	Ingested
		Expansion key	Ingested

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
