# Peer review of "An Ingested Orthodontic Wire Fragment: A Case Report"

_dentistry, 2016, doi:10.3390/dj4030024_

Round 1

Reviewer 1 Report

This case report is well written and addresses an important question.  I like to point out only a couple of details:

The Introduction section includes relevant background data. However, because of the accidental nature of these events, the references comprise mainly various case reports.  

In the Discussion, the authors write “When an object is accidentally lost irretrievably to the oropharyngeal area, radiographs are essential…”

All objects do not have radio opacity and can’t therefore be detected in radiographs. Instead of radiographs, I suggest terms like “Imaging studies” or “radiologic evaluation” because they cover a wider range of examinations, e.g.  ultra sound etc.  

Author Response

We are very grateful for your prompt and very supportive review of our paper to be published. We have noted your valid suggestion of amending the term "radiographs" on line 116 of the paper, and this has been suitably changed.

Reviewer 2 Report

The article is intersting and well written.

Author Response

We are very grateful for your prompt and fully supportive review of our paper to be published in its current form.

Reviewer 3 Report

The authors presented a case of accidental ingestion during routinary orthodontic visit. The patient underwent several radiographic exams showing the fragment ingested moving from the upper piriform recess till the rectum. The fragment was discharged with no specific medical complications. Authors suggest the importance of an immediate and subsequent management of this undesired clinical cases as already stated in the literature. 

Confirmation of known facts with no original contribution. Could be re-presented with a completed and updated litterature review of aspiration/ingestion in dental practice reports.

Author Response

Thank you for your prompt review of our paper.

We were obviously disappointed by your views, but have revised our paper by way of adding two further recent references to the literature.

In addition, we feel that our paper does add to the current available literature, clarifies published contemporary guidelines on the management of such incidents, and importantly, reminds all clinicians of the potential for this complication to occur.

We hope that you will reconsider our paper and accept it in its revised form, and endorse it in line with the views of the other reviewers who were fully supportive of the paper.

Yours sincerely

Round 2

Reviewer 3 Report

i still think that this paper should be much more improved in order to have a good appeal for the readers.

I suggested to make a short review of the literature with other cases and i see only two more ref. 

Author Response

Dear sir

Thank you for your comments and for giving us the opportunity to revise our paper.

A systematic review of literature is usually undertaken where data is weak or equivocal with the purpose of framing a research question or hypothesis.  This is very difficult where the "data" is derived from case reports or case series which are essentially reporting clinical observations with no quantitative or qualitative outcomes.  Nevertheless we have undertaken a literature review as requested, presented the results in a new Table, and added a further 13 references to this report.  We hope this manuscript is now acceptable and that you will look favourably on these changes.

With many thanks.

Yours sincerely

James Puryer

Round 3

Reviewer 3 Report

-

Author Response

Dear sir

Thank you for giving us opportunity to further revise this paper.

We feel that we have now fully addressed the 5 comments needed for this 'Minor Revision' and would be very grateful if you would now view these changes in a positive light and recommend publication of this paper.

Your sincerely